# Use of Radial Basis Function Network to Predict Optimum Calcium and Magnesium Levels in Seawater and Application of Pretreated Seawater by Biomineralization as Crucial Tools to Improve Copper Tailings Flocculation

**Grecia Villca** [1] (ID), **Dayana Arias** [1,2,*] (ID), **Ricardo Jeldres** [1] (ID), **Antonio Pánico** [3] (ID), **Mariella Rivas** [2] (ID) **and Luis A. Cisternas** [1,*] (ID)

1 Departamento de Ingeniería Química y Procesos de Minerales, Universidad de Antofagasta, Av. Universidad de Antofagasta, 02800 Antofagasta, Chile; greciahassel05@gmail.com (G.V.); ricardo.jeldres@uantof.cl (R.J.)

2 Departamento de Biotecnología, Facultad de Ciencias del Mar y Recursos Biológicos (FACIMAR), Universidad de Antofagasta, Av. Universidad de Antofagasta, 02800 Antofagasta, Chile; mariella.rivas@uantof.cl

3 Telematic University Pegaso, Piazza Trieste e Trento 48, 80132 Naples, Italy; antonio.panico@unipegaso.it

\* Correspondence: dayana.arias@uantof.cl (D.A.); luis.cisternas@uantof.cl (L.A.C.)

**Abstract:** The combined use of the Radial Basis Function Network (RBFN) model with pretreated seawater by biomineralization (BSw) was investigated as an approach to improve copper tailings flocculation for mining purposes. The RBFN was used to set the optimal ranges of $Ca^{2+}$ and $Mg^{2+}$ concentration at different Ph in artificial seawater to optimize the performance of the mine tailings sedimentation process. The RBFN was developed by considering $Ca^{2+}$ and $Mg^{2+}$ concentration as well as pH as input variables, and mine tailings settling rate (Sr) and residual water turbidity (T) as output variables. The optimal ranges of $Ca^{2+}$ and $Mg^{2+}$ concentration were found, respectively: (i) 169–338 and 0–130 mg·$L^{-1}$ at pH 9.3; (ii) 0–21 and 400–741 mg·$L^{-1}$ at pH 10.5; (iii) 377–418 and 703–849 mg·$L^{-1}$ at pH 11.5. The settling performance predicted by the RBFN was compared with that measured in raw seawater (Sw), chemically pretreated seawater (CHSw), BSw, and tap water (Tw). The results highlighted that the RBFN model is greatly useful to predict the settling performance in CHSw. On the other hand, the highest Sr values (i.e., 5.4, 5.7, and 5.4 m·$h^{-1}$) were reached independently of pH when BSw was used as a separation medium for the sedimentation process.

**Keywords:** calcium; magnesium; Radial Basis Function Network (RBFN); settling rate; turbidity; biomineralization; mine tailings; water quality

## 1. Introduction

Copper (Cu) is the main product of the Chilean mining industry. Cu is usually found in the form of sulfide ores and is processed using a pyrometallurgical method, which involves the following operations: (1) ore extraction, (2) mineral processing, (3) smelting, and (4) refining. Specifically, mineral processing includes comminution, the froth flotation process to recover Cu from sulfide ores, and dewatering to recover and recycle water from concentrates and tailings [1].

The progressive depletion of high-grade ores in the near future will constrain mining factories to process a higher amount of rock to keep stable the production of valuable minerals, thus,

requiring progressively higher volumes of water. Therefore, a further increase in water demand from the mining sector might be solely fulfilled by using seawater [2,3].

The use of raw seawater causes various operational problems [4]; for example, flotation operations affect bubble stability, activation of minerals, and pulp rheology, among others. There are several studies and reviews about the use of raw seawater directly in flotation processes [5–10]. A review of these studies is outside of the objective of this manuscript. However, another effect in mining operations includes low tailings thickening efficiency when operating under highly alkaline conditions. Specifically, $Ca^{2+}$ and $Mg^{2+}$ ions have a substantial effect on the shielding of electric charges [11]; hence, they can more intensely reduce the conformation of polyelectrolytes and lower flocculation efficiency. However, this depends on their concentration in solution, flocculant management, and mineralogy [12]. For example, Witham et al. [13] found that in calcite suspensions, $Ca^{2+}$ and $Mg^{2+}$ ions limited sedimentation rates at high polymer dosages. Moreover, their effect was hidden when the mixing conditions during flocculation were particularly intense. On the other hand, Lee et al. [14] pointed out that low levels of $Ca^{2+}$ and $Mg^{2+}$ could improve polymer adsorption and increase the size of kaolinite flocs when anionic polyacrylamides are added as flocculants.

The interest in the copper industry is to be able to operate the concentration stages at pH above 10.5, to promote the pyrite depression in flotation stages. However, this alkaline condition induces the formation of $Mg^{2+}$ solid complexes, which are insoluble in the medium. A recent study of Ramos et al. [15] showed that these complexes disable the active sites of the flocculant, thus, reducing its ability to form hydrogen bonds and, consequently, causing a drop in the density and size of flocs. The occurrence of such events is responsible for the low sedimentation rates at pH higher than 10.3. Recently, Jeldres et al. [16] reduced the $Mg^{2+}$ concentration in seawater by adding lime and filtering the solid complexes. Such water was evaluated as a medium in tailings flocculation, finding out that this method could improve the performance of thickeners. The authors obtained a considerable increase in the sedimentation rate of mineral tailings at pH 11, reaching values that even exceeded the performance obtained at natural pH, which represents the most common current operating condition.

Based on this background, it is evident that divalent cations in seawater affect the performance of the sedimentation process. Therefore, partial desalination of seawater aimed at reducing the concentration of $Ca^{2+}$ and $Mg^{2+}$ could be useful to increase settling efficiency. Cruz et al. [17] exhibited that lowering the concentration of $Ca^{2+}$ and $Mg^{2+}$ ions by using carbon dioxide gas and a sodium hydroxide solution, improved the flocculation of clay-based tailings. Moreover, Arias et al. [18] proposed a biotechnological treatment to remove divalent ions and obtain pretreated seawater by biomineralization by using a fluidized bed bioreactor (FBB) inoculated with halotolerant ureolytic strain *Bacillus subtilis* LN8B. Specifically, the process used to obtain pretreated seawater by biomineralization is conducted by ureolytic bacteria capable of forming different crystalline elements from seawater through microbial-induced carbonate precipitation (MICP) [18]. Both studies have shown substantial improvements in tailings flocculation; however, no study has quantified the impact of the synergy concentration of these divalent ions under highly alkaline conditions. Such information is essential to know the optimal concentration ranges for both ions, useful for optimizing seawater treatment practice.

Finally, in this paper, a radially based neural network method (RBFN) was used to find the optimal $Ca^{2+}$ and $Mg^{2+}$ concentration ranges to obtain the highest sedimentation rate and lowest residual turbidity. The RBFN method consists of an input layer, hidden layer, and an output layer, with the activation function of the hidden units being radial basis functions. To model and optimize the process with a reasonable number of experiments, a proper experimental plan was designed according to the RBFN method [19]. Furthermore, the desalination of seawater was carried out biologically and chemically through the replication of research from Arias et al. [18] and Cruz et al. [1], respectively.

## 2. Materials and Methods

### 2.1. Artificial Mine Tailings

The artificial mining tailings used for sedimentation process tests were prepared by mixing 80% w/w quartz, ($SiO_2$, Donde Capo Industry, Ñuñoa, Chile, density = 2.67 g·cm$^{-3}$) with 20% w/w kaolinite ($Al_2Si_2O_5(OH)_4$, Ward Science, Henrietta, NY, USA, density = 2.50 g·cm$^{-3}$). Both minerals were sieved with ASTM sieves in sizes lower than 75 μm. The composition of the mixture was analyzed by X-Ray Diffraction (XRD) using a diffractometer (Siemens D5000, Madison, WI, USA), and the relative abundances are reported in Table 1. Quartz purity was 95.3%, whereas kaolinite was 98.4%.

**Table 1.** Composition of the mixture of quartz and kaolinite used to simulate the artificial mine tailings.

| Sample | Crystalline Phase | Chemical Composition | Abundance (%) |
|---|---|---|---|
| Quartz | Quartz | $SiO_2$ | 95.3 |
| | Albite | $NaAlSi_3O_8$ | 0.7 |
| | orthoclase | $KAlSi_3O_8$ | 1.3 |
| | microcline | $KAlSi_3O_8$ | 2.6 |
| Kaolinite | kaolinite | $Al_2Si_2O_5(OH)_4$ | 98.4 |
| | Quartz | $SiO_2$ | 1.6 |

### 2.2. Separation Medium

#### 2.2.1. Artificial Seawater

Artificial seawater (ASw) was obtained by adding different chemical reagents to distilled water following the composition shown in Table 2 and according to Mobin and Sharman [20]. Ultra-high purity chemical reagents were used (Merck, Darmstadt, Germany). Different concentrations of $Ca^{2+}$ and $Mg^{2+}$ were set by adequately varying the dosage of $CaCl_2$ and $MgCl_2 \times 6H_2O$, respectively.

**Table 2.** Composition of artificial seawater, according to Mobin et al. 2011.

| Component | Concentration (g·L$^{-1}$) |
|---|---|
| NaCl | 24.53 |
| $MgCl_2·6H_2O$ | 11.10 |
| $Na_2SO_4$ | 4.09 |
| $CaCl_2$ | 1.16 |
| KCl | 0.69 |
| $NaHCO_3$ | 0.20 |
| KBr | 0.10 |
| $H_3BO_3$ | 0.03 |

#### 2.2.2. Pretreated Seawater by Biomineralization

To obtain pretreated seawater by biomineralization (BSw), the concentrations in raw seawater of $Ca^{2+}$ and $Mg^{2+}$ were reduced through precipitation by the ureolytic bacterial strain *B. subtilis* LN8B, isolated from the hypersaline lagoon of the desert of San Pedro de Atacama, Chile [21]. BSw was obtained from bioreactors filled with polyvinyl alcohol and alginate beads (PVA-Alg beads) with entrapped cells of *B. subtilis* LN8B, according to the experimental system and procedure as described by Arias et al. [18].

#### 2.2.3. Chemically Pretreated Seawater

Raw seawater was chemically pretreated (CHSw) to partially remove $Ca^{2+}$ and $Mg^{2+}$, using the procedure as described by Cruz et al. [1] with some changes as follows: 1.4 L of raw seawater was alkalinized by adding 10 M NaOH to reach a pH of 10.5. Afterward, $CO_2$ gas was flushed at

3 L min$^{-1}$ for 6 min, and simultaneously, the solution of NaOH was added for keeping the pH at 11. Such treatment resulted in a white-colored solution because of the presence of microcrystals of $Ca^{2+}$- and $Mg^{2+}$-based salts, which were removed by a vacuum filtration system equipped with a pump (Vacuubrand industry, Wertheim, Germany) and 12 μm filters (Ahlstrom-Munksjö, Helsinki, Finland).

### 2.2.4. Tap Water and Raw Seawater

Tap water (Tw), coming from a reverse osmosis process containing average concentrations of $Ca^{2+}$ and $Mg^{2+}$ of 74 ± 11 and 23 ± 5 mg·L$^{-1}$, respectively, was collected from the aqueduct supplying water to the city of Antofagasta. In contrast, raw seawater (Sw) was obtained from San Jorge Bay of Antofagasta, Chile. Prior to use, raw seawater was filtrated progressively with filters of 10, 5, and 1 μm porous size, respectively, to remove the main suspended impurities. The average $Ca^{2+}$ and $Mg^{2+}$ concentrations were detected equal to 444 ± 8 and 1283 ± 13 mg·L$^{-1}$, respectively.

### 2.3. Sedimentation Tests

Sedimentation tests were carried out with artificial tailings 15% w/w suspended in 5 different aqueous media, as follows: ASw, BSw, CHSw, Tw, and Sw. The adopted procedure is schematically summarized in Figure 1:

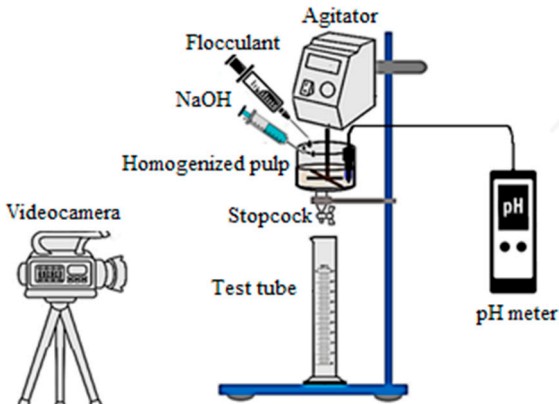

**Figure 1.** Experimental apparatus.

- Step 1: the mine tailings are added to the aqueous medium by filling a beaker and homogenized by mechanical mixing at 500 rpm for 10 min;
- Step 2: pH is regulated at a specific set point (i.e., 9.3, 10.5, 11.5) by adding different volumes of 10 M NaOH solution;
- Step 3: the anionic polyacrylamide flocculant SNF 60430 obtained from Centinela copper mine (Antofagasta, Chile) is added to the aqueous suspension according to a dosage of 20 g·ton$^{-1}$ and the mechanical mixer is turned down to 170 rpm and kept constant for 30 s;
- Step 4: the aqueous suspension filling the beaker is poured into a graduated cylinder and kept in static conditions for 1 h;
- Step 5: the volume of supernatant is collected after one hour of sedimentation process and used to measure the turbidity T (FNU), while for the solid fraction (*Cp*), it is measured using Equation (1), where $m_t$ is the artificial tailing mass, $\rho_w$ is water density (1g·cm$^{-3}$), $Vi$ is the initial total volume of water needed to form the pulp, and $Vs$ is the supernatant volume. The percentage of water recovery (% *WR*) is calculated with Equation (2).

$$Cp = \frac{m_t}{\rho_w\,(Vi - Vs) + m_t} \tag{1}$$

$$\% \, WR = \frac{Vs}{Vi} 100\% \tag{2}$$

A video camera recorded the sedimentation process that took place in the graduated cylinder. For each sedimentation test, the related video was processed to calculate the following parameter: (i) the mine tailings settling rate, $Sr$ (m·h$^{-1}$), described by Equation (3).

$$Sr = \frac{\Delta h}{\Delta t} \tag{3}$$

where $h$ (m) is the height of the clarified surface water and $t$ (h) is the time taken by the clarified surface water to reach the minimum height in the graduated cylinder.

### 2.4. Experimental Setting

To maximize the efficiency of the sedimentation process of mine tailings in ASw, it was necessary to find the optimal concentrations of $Ca^{2+}$ and $Mg^{2+}$ ions by varying the pH. This study tested operating conditions where the concentration of the $Ca^{2+}$ and $Mg^{2+}$ ions was set lower than that typical in seawater. The value of pH was set higher than 9 due to the flotation process operating condition (the flotation process precedes the sedimentation process). The value of pH 9.3 was used as the initial value because of BSw (the biomineralization process pH is set to 9.3). The three-level full factorial experimental plan ($3^3$ experimental tests) was elaborated as reported in Table 3: $Ca^{2+}$ concentration ranging between 0 and 418.89 mg·L$^{-1}$; $Mg^{2+}$ concentration ranging between 0 and 1327; finally, pH ranging between 9.3 and 11.5. A third value, intermediate between the extremes of each interval, was used to complete the $3 \times 3$ matrix of experiments.

**Table 3.** Parameters and relative values used for setting the experimental plan.

| Parameter | Symbol | Level | | |
|---|---|---|---|---|
| | | **Low** | **Intermediate** | **High** |
| $Ca^{2+}$ concentration (mg·L$^{-1}$) | $Ca^{2+}$ | 0 | 209.445 | 418.89 |
| $Mg^{2+}$ concentration (mg·L$^{-1}$) | $Mg^{2+}$ | 0 | 663.5 | 1327 |
| pH | pH | 9.3 | 10.5 | 11.5 |

Finally, a Radial Basis Function Network (RBFN) model [22] was used by considering a three-level factorial design with three independent input variables: $Ca^{2+}$ concentration, $Mg^{2+}$ concentration, and pH; two response output variables: mine tailing settling rate (Sr) and supernatant turbidity (T). Numerical simulations were performed using the software of numerical computing MATLAB (version 2009a, MathWorks, Natick, MA, USA).

### 2.5. Physical and Chemical Analyses

$Ca^{2+}$ and $Mg^{2+}$ concentrations were measured by using Merck® equipment (cat. No. 1.14815.0001 and 1.00815.0001, respectively; Darmstadt, Germany) and corroborated by Atomic Absorption spectrophotometry analyses, whereas pH was measured by using a digital pH meter model HI5222 (Hanna instruments, Woonsocket, RI, USA). Measures of turbidity were performed with turbidimeter model H198713 (Hanna instruments, Woonsocket, RI, USA).

### 2.6. Statistical Analysis

The average values of the experimental data—Sr, T, Cp, % WR, and the NaOH consumption—are presented as a function of two factors: type of water and pH. The standard deviations of each set of experiments are represented in the corresponding figure (as bars). Comparisons of the means of Sr, T, Cp, % WR, and the NaOH consumption were performed to identify statistical significance as a function of water type and pH. Minitab software version 19.1 (State College, PA, USA) through ANOVA general

linear model tests, was used to conduct the statistical analysis. Comparisons between water types and pH values were evaluated with the Tukey test. In all investigated cases, the level of significance was set $p < 0.05$.

### 2.7. Radial Basis Function Network (RBFN)

Neural networks can represent non-linear assignments of multiple inputs to one or more outputs. These can also be applied to regression problems. The neural network is capable of mapping input variables to continuous values. An important class of neural networks is RBFN. The RBFN consists of an input layer, a hidden layer (Radial Basis Layer), and an output layer (linear layer). The structure of the RBFN is presented in Figure 2, where the inputs are represented by $p^1$ and the outputs by $a^2$.

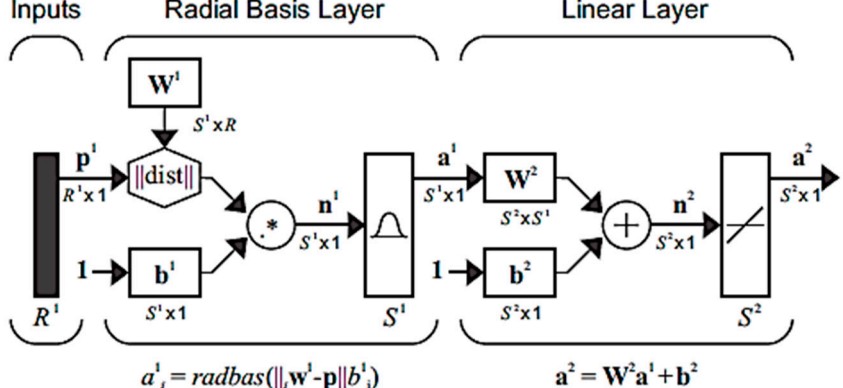

**Figure 2.** Radial Basis Function Network (RBFN) structure.

In the Radial Basis Layer, the distance between the input vector $p_i$ and the rows of the weight $w_i^1$ are calculated, and then multiplied by the bias $b_i$. The net input for the neuron $i$ ($n_i$) in the Radial Basis Layer is calculated by Equation (4):

$$n_i^1 = \|p_i - w_i^1\| b_i \tag{4}$$

where $b_i$ is the bias, which is related with standard deviation $\sigma$ with Equation (5).

$$b_i = \frac{1}{\sigma \sqrt{2}} \tag{5}$$

The transfer function used in the Radial Basis Layer or hidden layer is a Gaussian function, which is commonly used in the neural network community. Equation (6) and Figure 3 represent it.

$$a^1 = exp\left(-\left(n_i^1\right)^2\right) \tag{6}$$

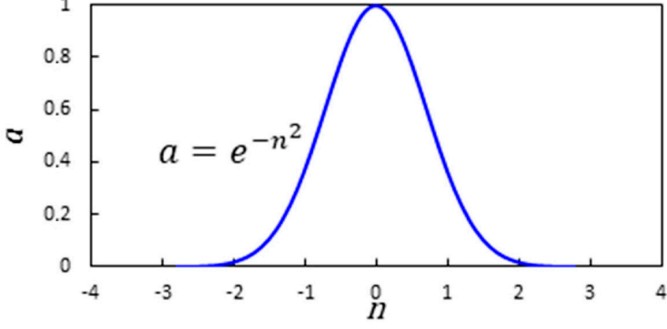

**Figure 3.** Gaussian Basis function.

Finally, the linear layer or the output layer is represented by Equation (7):

$$a^2 = w^2 a^1 + b^2 \tag{7}$$

The performance of the RBFN is evaluated using the mean squared error (MSE). The next equations describe the sequence to calculate the MSE using the RBFN. Let us consider the following training points, in Equation (8):

$$\{p_1, t_1\}, \{p_1, t_1\}, \ldots\ldots, \{p_Q, t_Q\} \tag{8}$$

The neuron in the hidden layer is calculated by Equations (9) and (10):

$$n_i{}^1 = \|p_i - w_i{}^1\| b_i{}^1 \tag{9}$$

$$a^1 = exp\left(-\left(n_i{}^1\right)^2\right) \tag{10}$$

Grouping terms, the following points are obtained in Equation (11):

$$\left\{a_1^1, t_1\right\}, \left\{a_2^1, t_2\right\}, \ldots\ldots, \left\{a_Q^1, t_Q\right\} \tag{11}$$

Then, the response $a^2$ is determined in the output layer through Equation (12):

$$a^2 = w^2 a^1 + b^2 \tag{12}$$

Finally, the MSE performance index is determined using Equation (13) for the training set [23]:

$$F(x) = \sum_{q=1}^{Q} \left(t_q - a_q^2\right)^T \left(t_q - a_q^2\right) \tag{13}$$

The newrb function of MATLAB was used for modeling the sedimentation rate and turbidity using the RBFN. Two RBFNs were built, one for the sedimentation rate and another for the turbidity. Both networks have three input variables ($Ca^{2+}$ and $Mg^{2+}$ concentrations and pH). The newrb function works with the following code: net = newrb ($p$, $t$, GOAL, SPREAD). The $p$ matrix represents the input variables and $t$ the output variables. GOAL is the target error and SPREAD represents the spread width, which can take different values that will influence the MSE. The network is developed by adding one neuron at a time, and each calculates the MSE. In the iterative process, the MSE value is compared with the GOAL, and if the MSM is equal to or less than GOAL, then, the process converges; otherwise, another neuron is added and so on [24]. This iterative process is performed for various values of SPREAD.

It is worth indicating that the RBFN was selected because a recent study shows that the RBFN performs better than multi-layer perceptron networks when it is applied to response surface methodology [25]. Another advantage of the RBFN over other neural networks is that during RBFN learning, the inputs are fed directly from the hidden layer without any weight, and the weights are only manifested between the hidden and the output layer. Such weights are modified depending on the error. Thus, the RBFN requires a much shorter learning time compared to multi-layer feedback neural networks used in other practical applications, and therefore, the convergence time is also considerably short [26].

## 3. Results and Discussions

### 3.1. Sedimentation Tests in Artificial Seawater

Table 4 summarizes the results in terms of Sr, Cp, % WR, and T of the 27 experimental sedimentation tests in artificial seawater (ASw). The results were obtained combining factorially the three selected

parameters (i.e., $Ca^{2+}$, $Mg^{2+}$, and pH), as indicated in the first three columns of Table 4. The ionic strength of ASw (I) is also incorporated. Note that when the concentration is 0 mg·L$^{-1}$ for both $Ca^{2+}$ and $Mg^{2+}$ ions, the ionic strength is 0.52 mol·L$^{-1}$. This condition is due to the composition of ASw, which has other salts described in Table 2.

**Table 4.** Results of sedimentation tests in ASw.

| Independent Parameters | | | | Response Parameters | | | |
|---|---|---|---|---|---|---|---|
| $Ca^{2+}$ (mg·L$^{-1}$) | $Mg^{2+}$ (mg·L$^{-1}$) | pH | I (mol·L$^{-1}$) | Sr (m·h$^{-1}$) | Cp (w/w) | % WR (%) | T (FNU) |
| 418.89 | 1327 | 9.3 | 0.71 | 7.368 | 0.475 | 87.4 | 77.8 |
| 418.89 | 1327 | 10.5 | 0.71 | 6.006 | 0.427 | 84.7 | 47.2 |
| 418.89 | 1327 | 11.5 | 0.71 | 0.403 | 0.255 | 66.6 | 38.6 |
| 418.89 | 663.5 | 9.3 | 0.63 | 3.266 | 0.454 | 86.3 | 88 |
| 418.89 | 663.5 | 10.5 | 0.63 | 4.495 | 0.448 | 86.0 | 98.9 |
| 418.89 | 663.5 | 11.5 | 0.63 | 4.293 | 0.400 | 82.9 | 48.7 |
| 418.89 | 0 | 9.3 | 0.55 | 4.535 | 0.481 | 87.7 | 106 |
| 418.89 | 0 | 10.5 | 0.55 | 4.289 | 0.481 | 87.7 | 102 |
| 418.89 | 0 | 11.5 | 0.55 | 7.352 | 0.462 | 86.7 | 117 |
| 209.4 | 1327 | 9.3 | 0.70 | 4.703 | 0.476 | 87.5 | 132 |
| 209.4 | 1327 | 10.5 | 0.70 | 6.451 | 0.438 | 85.4 | 95 |
| 209.4 | 1327 | 11.5 | 0.70 | 0.579 | 0.271 | 69.4 | 13.8 |
| 209.4 | 663.5 | 9.3 | 0.62 | 3.505 | 0.462 | 86.7 | 84.5 |
| 209.4 | 663.5 | 10.5 | 0.62 | 4.059 | 0.419 | 84.2 | 26.2 |
| 209.4 | 663.5 | 11.5 | 0.62 | 2.137 | 0.391 | 82.3 | 15.2 |
| 209.4 | 0 | 9.3 | 0.54 | 7.804 | 0.477 | 87.5 | 120 |
| 209.4 | 0 | 10.5 | 0.54 | 7.515 | 0.455 | 86.4 | 151 |
| 209.4 | 0 | 11.5 | 0.54 | 6.047 | 0.475 | 87.4 | 128 |
| 0 | 1327 | 9.3 | 0.68 | 4.206 | 0.485 | 87.9 | 13.8 |
| 0 | 1327 | 10.5 | 0.68 | 4.249 | 0.451 | 86.2 | 137 |
| 0 | 1327 | 11.5 | 0.68 | 0.639 | 0.279 | 70.6 | 9.72 |
| 0 | 663.5 | 9.3 | 0.60 | 3.367 | 0.467 | 87.0 | 66.7 |
| 0 | 663.5 | 10.5 | 0.60 | 6.887 | 0.416 | 84.1 | 39.3 |
| 0 | 663.5 | 11.5 | 0.60 | 2.108 | 0.381 | 81.6 | 14.5 |
| 0 | 0 | 9.3 | 0.52 | 3.418 | 0.488 | 88.1 | 238 |
| 0 | 0 | 10.5 | 0.52 | 3.444 | 0.486 | 88.0 | 218 |
| 0 | 0 | 11.5 | 0.52 | 2.965 | 0.467 | 87.0 | 85 |

Figure 4 shows the decrease in Sr and T for high $Mg^{2+}$ concentrations (1327 mg·L$^{-1}$) at pH 10.5–11.5, which demonstrates the effect of pH and $Mg^{2+}$ in Sr and T. Furthermore, it is noted that at pH 10.5, the sedimentation rates are the highest and at pH 11.5, the turbidity values are the lowest.

With this explanation, it is observed that the graphs in Figure 4 do not show an ordered and proportional sequence, and this result can be due to the non-monotonous behavior of such ions in the progress of the process [27]. Therefore, the model capable of fitting the data is complex. An adjustment was made with the Multiple Regression method by using Minitab software version 19.1 (State College, PA, USA) and two equations governing the behavior of the settling rate (Sr) and turbidity (T) were obtained. However, the adjusted $R^2$ of those equations resulted in being 14% and 35%, respectively. Several multiphase systems do not follow a second-order polynomial behavior like what was observed here. The immediate consequence is incorrect optimization. In these cases, the most popular alternative is to use ANNs [28]. For this reason, the RBFN method was chosen, and it was advantageous to show more comprehensively the results (see Section 3.2).

To better understand the results of Sr and T obtained from the sedimentation tests, in Figure 5, such response parameters are studied in the function of the solution ionic strength (I). It is well known that I depends on the dissociation degrees of salts in water and the concentration of ions and their valence [29]; therefore, the nine combinations of $Ca^{2+}$ and $Mg^{2+}$ concentrations resulted in nine different values of I, as indicated in the *X*-axis of the graphs in Figure 5.

According to Mpofu et al. [30], an increase in ionic strength can reduce the repulsive force barrier between particles by compressing the thickness of the double layer, thus, enabling particle aggregation and consequently, improving sedimentation process performance. As all ions dissolved in the water contribute to the value of ionic strength and not only those divalent investigated in this study, it is

crucial to know the complete chemical composition of water. The experimental results show that at pH 9.3, 10.5, and 11.5, high settling rate and a low residual turbidity are achievable with the following conditions: (i) at pH = 9.3 with I = 0.68 mol·L$^{-1}$; (ii) at pH = 10.5 with I = 0.60 mol·L$^{-1}$; (iii) at pH = 11.5 with I = 0.60 mol·L$^{-1}$.

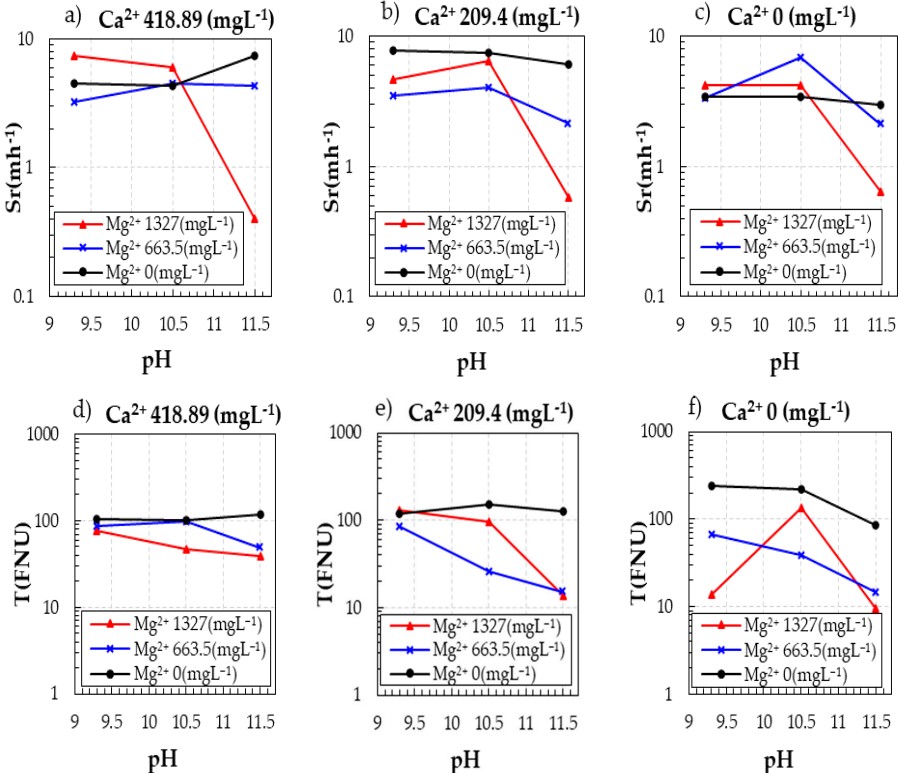

**Figure 4.** Trends of settling rate (Sr) and turbidity (T) by varying Ca$^{2+}$ and Mg$^{2+}$ concentrations as well as pH. (**a**) Sr at 418 mg·L$^{-1}$ of Ca$^{2+}$, (**b**) Sr at 209.4 mg·L$^{-1}$ of Ca$^{2+}$, (**c**) Sr at 0 mg·L$^{-1}$ of Ca$^{2+}$, (**d**) T at 418 mg·L$^{-1}$ of Ca$^{2+}$, (**e**) T at 209.4 mg·L$^{-1}$ of Ca$^{2+}$, (**f**) T at 0 mg·L$^{-1}$ of Ca$^{2+}$.

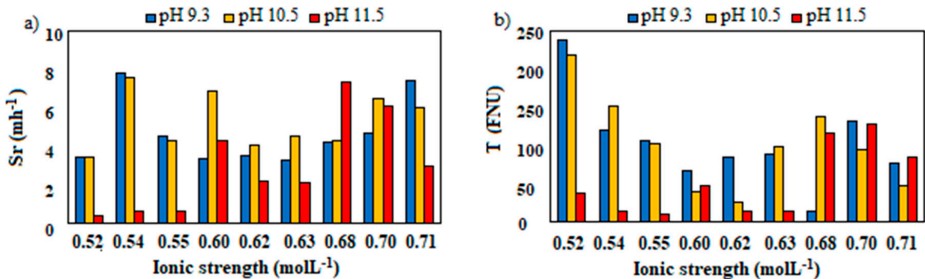

**Figure 5.** Mine tailings settling rate and turbidity as a function of the ionic strength at pH 9.3, 10.5, and 11.5. (**a**) Mine tailings settling rate and (**b**) turbidity as a function of Ionic strength at different pH values.

Figure 6 shows the trends of % WR, Cp, and NaOH, varying the concentrations of Ca$^{2+}$ and Mg$^{2+}$ and the pH according to the selected ranges. It can be seen that the Ca$^{2+}$ concentration minimally affects the performance of the sedimentation process of the three selected response parameters. A different conclusion is reached when the effect of Mg$^{2+}$ concentration is taken into account: when the Mg$^{2+}$ concentration increases, % WR, and Cp decrease, while NaOH consumption increases. For example, at the highest concentration of Mg$^{2+}$ (i.e., 1327 mg·L$^{-1}$), % WR decreases approximately by 20%, Cp decreases from 0.5 to 0.3, and NaOH consumption is 5–8 times higher in the pH range 10.5–11.5, than in the range 9.3–10.5. This result is due to the strong buffer effect of seawater in a zone that

coincides with the formation of solid magnesium species [31,32]. These results agree with those reported by Ramos et al. [15], who stated that the presence of magnesium precipitates deactivated the functional groups of anionic polyacrylamides. The flocculant lost selectivity, thus, being unable to bridge particles that led to large aggregates. However, in the pH range considered in this study, calcium is mainly dissolved in solution, so its impact on flocculant performance is not as relevant as magnesium.

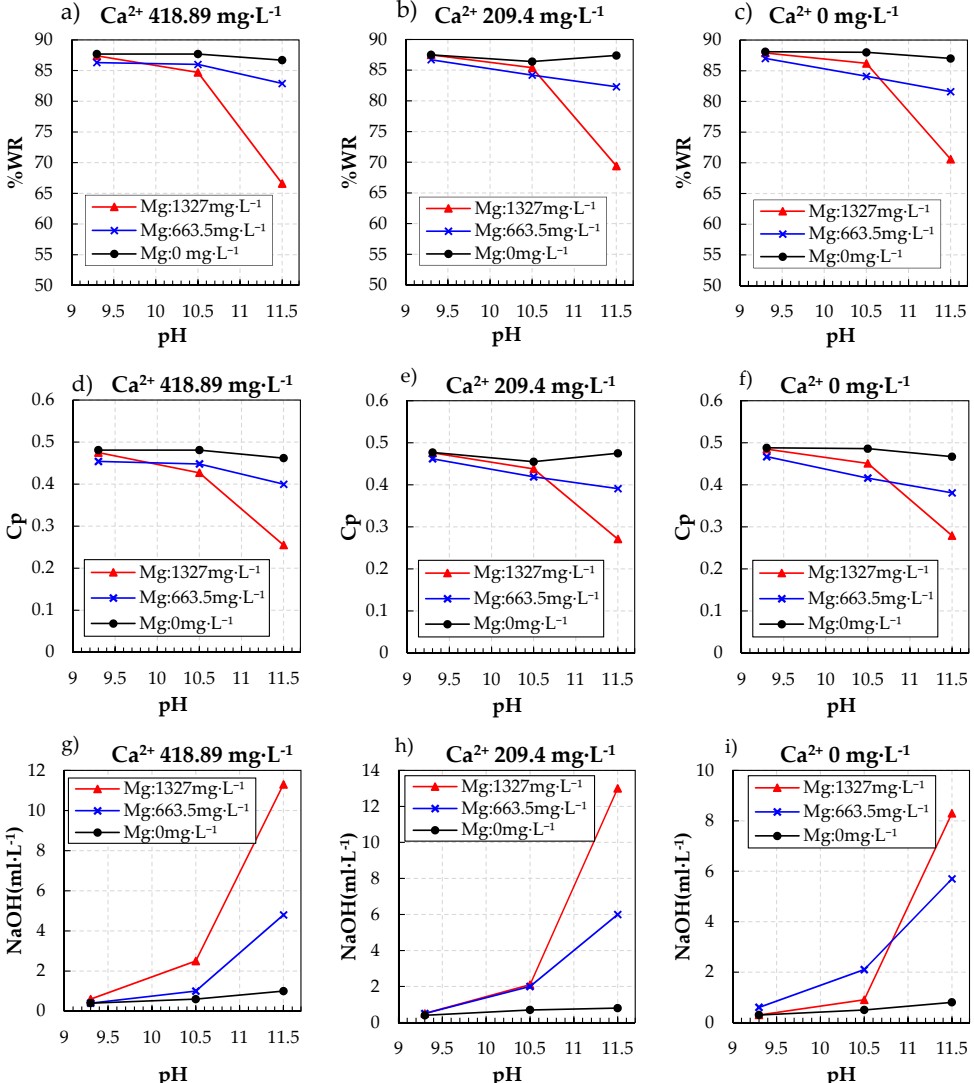

**Figure 6.** Trends of solid fraction (Cp), water recovery (% WR), and NaOH solution consumption by varying Ca$^{2+}$ and Mg$^{2+}$ concentrations as well as pH. (**a**) % WR at 418.89 mg·L$^{-1}$ of Ca$^{2+}$, (**b**) %WR at 209.4 mg·L$^{-1}$ of Ca$^{2+}$, (**c**) %WR at 0 mg·L$^{-1}$ of Ca$^{2+}$, (**d**) Cp at 418.89 mg·L$^{-1}$ of Ca$^{2+}$, (**e**) Cp at 209.4 mg·L$^{-1}$ of Ca$^{2+}$, (**f**) Cp at 0 mg·L$^{-1}$ of Ca$^{2+}$, (**g**) NaOH solution consumption at 418.89 mg·L$^{-1}$ of Ca$^{2+}$, (**h**) NaOH solution consumption at 209.4 mg·L$^{-1}$ of Ca$^{2+}$, (**i**) NaOH solution consumption at 0 mg·L$^{-1}$ of Ca$^{2+}$.

### 3.2. Analysis of the Radial Basis Function Network (RBFN) Model

Artificial neural networks allow experimental data to be processed, generating a predictive dataset (RBFN output data) which is suitable for achieving the optimal solution of non-linear equations [33]. After calibration, the RBFN model turns to be a potent tool capable of predicting, with accuracy, the output response variables from any combination of input values (independent variables). Recall that the RBFN structure has an input layer, a hidden layer, and an output layer. Within our case, the RBFN

has three input variables ($Ca^{2+}$ concentration, $Mg^{2+}$ concentration, and pH) and two output variables (Sedimentation Rate, Sr, and Turbidity, T). The MATLAB computer program was used to find the optimal number of neurons and the width (w) described in Section 2.7. The objective function was to minimize MSE [34].

In data processing with RBFN, 27 data points were processed for settling rate and turbidity. In the first step, the three input variables were normalized to values in the range [0, 1] to facilitate the calculation and eliminate the influence of dimension. In the second step, the function newrb (p,t, GOAL, SPREAD) of MATLAB was applied with a GOAL of 0.05. Several values of SPREAD were employed and SPREAD values of 2.49 and 0.4 were found for the settling rate and turbidity, respectively. For these SPREAD values, 20 and 25 neurons were obtained for settling rate and turbidity networks. Figure 7 provides the RBFN structures obtained. Table 5 presents weight **w** and bias **b** values for hidden ($w^1$, $b^1$) and output layer ($w^2$, $b^2$) for settling rate and turbidity, respectively.

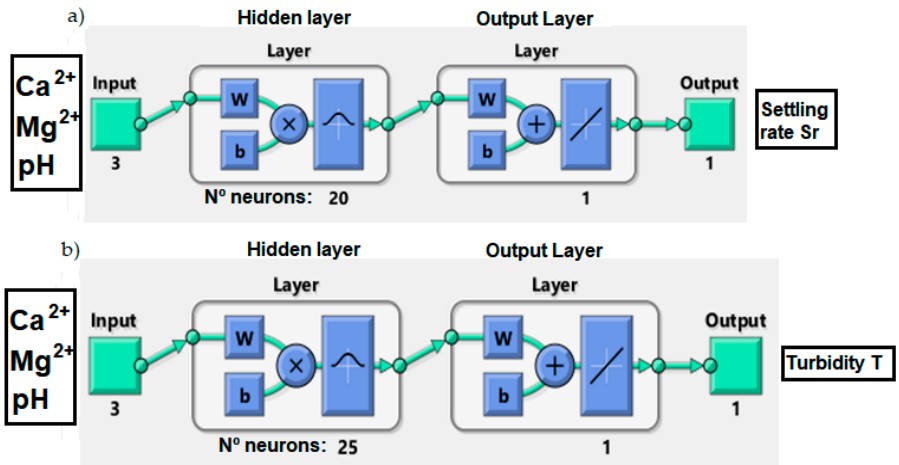

**Figure 7.** MATLAB structure for the RBFN (**a**) settling rate and (**b**) turbidity.

**Table 5.** Settling rate and turbidity weights (w) and bias (b) values from processing data by MATLAB.

| Settling Rate Sr | | | | | | Turbidity T | | | | | |
|---|---|---|---|---|---|---|---|---|---|---|---|
| $w^1$ | | | $w^{2T}$ | $b^1$ | $b^2$ | $w^1$ | | | $w^{2T}$ | $b^1$ | $b^2$ |
| 1 | 0 | 0 | 160,320.7 | 0.3344 | −53.425 | 0 | 0 | 0.5455 | 174.2 | 2.0814 | −2.622 |
| 0.4999 | 1 | 1 | 583,055.1 | | | 1 | 0 | 0.5455 | −10.8 | | |
| 1 | 0.5 | 0.5455 | 400,102.9 | | | 0 | 0 | 0 | 198.1 | | |
| 1 | 1 | 1 | −227,975.1 | | | 0.4999 | 1 | 0 | 123.2 | | |
| 0 | 0.5 | 1 | 386,413.9 | | | 0 | 1 | 0.5455 | 196.2 | | |
| 0 | 0 | 1 | −126,579.5 | | | 0.4999 | 0 | 1 | 71.7 | | |
| 0 | 1 | 1 | −245,670.7 | | | 1 | 0.5 | 0 | 8.9 | | |
| 0.4999 | 0 | 0.5455 | 232,292.3 | | | 0 | 0.5 | 0.5455 | −88.9 | | |
| 1 | 0.5 | 1 | 252,277.5 | | | 0 | 1 | 0 | −76.5 | | |
| 0.4999 | 0 | 1 | 256,532.2 | | | 0 | 1 | 1 | −67.4 | | |
| 0 | 1 | 0 | 17,756.8 | | | 1 | 1 | 1 | 43.5 | | |
| 0.4999 | 1 | 0.5455 | −141,222.4 | | | 1 | 0 | 0 | 92.4 | | |
| 1 | 0 | 0.5455 | −380,104.8 | | | 1 | 0 | 1 | 90.2 | | |
| 0 | 0 | 0.5455 | −107,365.3 | | | 1 | 0.5 | 0.5455 | 99.9 | | |
| 1 | 0.5 | 0 | −223,612.6 | | | 0.4999 | 0.5 | 1 | −14.4 | | |
| 1 | 1 | 0.5455 | −83,433.9 | | | 0.4999 | 0.5 | 0.5455 | −75.0 | | |
| 0 | 0.5 | 0.5455 | 64,578.8 | | | 0.4999 | 0 | 0.5455 | 56.4 | | |
| 0.4999 | 0.5 | 1 | −870,233.9 | | | 1 | 0.5 | 1 | −28.2 | | |
| 0.4999 | 0 | 0 | −39,731.5 | | | 0.4999 | 1 | 0.5455 | 33.6 | | |
| 1 | 1 | 0 | 92,488.7 | | | 0 | 0.5 | 1 | 30.6 | | |
| | | | | | | 1 | 1 | 0 | 28.5 | | |
| | | | | | | 1 | 1 | 0.5455 | −23.2 | | |
| | | | | | | 0 | 0 | 1 | −12.6 | | |
| | | | | | | 0.4999 | 0.5 | 0 | 19.1 | | |
| | | | | | | 0.4999 | 0 | 0 | −10.6 | | |

According to values shown in Table 6, the statistical parameter $R^2$ resulting from the calibration process was 0.98 and 0.99 for Sr and T, respectively. These values, coupled with other statistical parameters reported in Table 6, corroborate the reliability of the RBFN model in simulating results from the sedimentation process. Figure 8 shows graphically the fitting between experimental data and those predicted by the RBFN model.

**Table 6.** Statistical parameters resulting from the calibration of the Radial Basis Function Network (RBFN) model.

| Statistical Parameters | Value | |
| --- | --- | --- |
| | **Sr Prediction** | **T Prediction** |
| Mean relative error | 0.1104 | 0.00597 |
| Mean square error | 0.1638 | 0.29917 |
| Root mean square | 0.4047 | 0.54697 |
| R squared | 0.9809 | 0.99996 |
| Adjusted R squared | 0.9774 | 0.99995 |

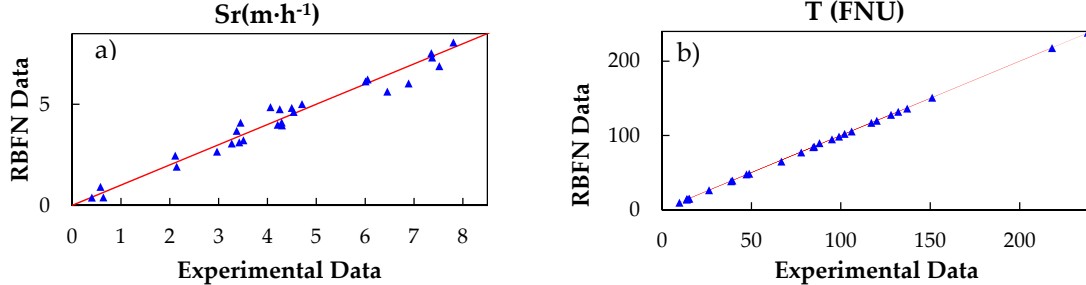

**Figure 8.** Comparison between experimental data and predicted data by the Radial Basis Function Network (RBFN) model for (**a**) Settling rate (Sr), (**b**) Turbidity (T).

### 3.3. Radial Basis Function Network (RBFN) Validation

There are different methods of network validation. These methods mainly depend on the amount of data available. According to Kopal et al. [35], three ways of validation are known. For a small dataset, the following validation method is adequate. The idea of this method is to train the network several times using the following methodology: from a set composed of a number $n$ of samples or points, one of them is omitted, and the neural network is trained; then, the next point is omitted, and the neural network is re-trained; therefore, one point at a time is removed and the network is trained. This process is iteratively performed as long as $n$ subsets of response data are generated. One of them is left for validation; therefore, $n - 1$ sets of responses are used to calibrate the model, and the remaining one is used for its validation.

The validation results are displayed in Figure 9, where the calculated values (average data) are compared with the last 27th set of data (validation). According to the previously described procedure, the validation for the Settling rate (Sr) and Turbidity (T) have adjusted $R^2$ of 0.90 and 0.92, respectively.

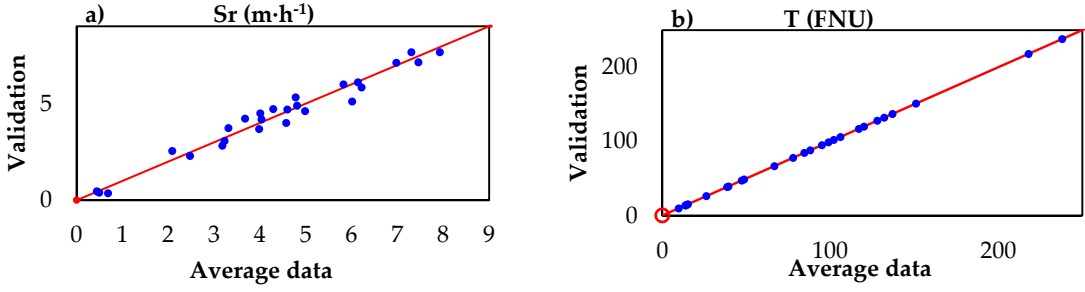

**Figure 9.** Comparison between validation and average data in the validation process for (**a**) Settling rate (Sr), (**b**) Turbidity (T).

### 3.4. Calcium and Magnesium Influence in Settling Rate (Sr) and Turbidity (T)

From the 3D surfaces plotted for different pH values (i.e., 9.3, 10.5, and 11.5) in Figure 10, it can be visibly noticed the variations of Sr (Figure 10a,c,e) and T (Figure 10b,d,f), predicted by the RBNF model as functions of $Ca^{2+}$ and $Mg^{2+}$ concentrations. The optimal range of the three independent variables can be found by setting as targets the maximization of tailings settling rate and the minimization of the supernatant turbidity.

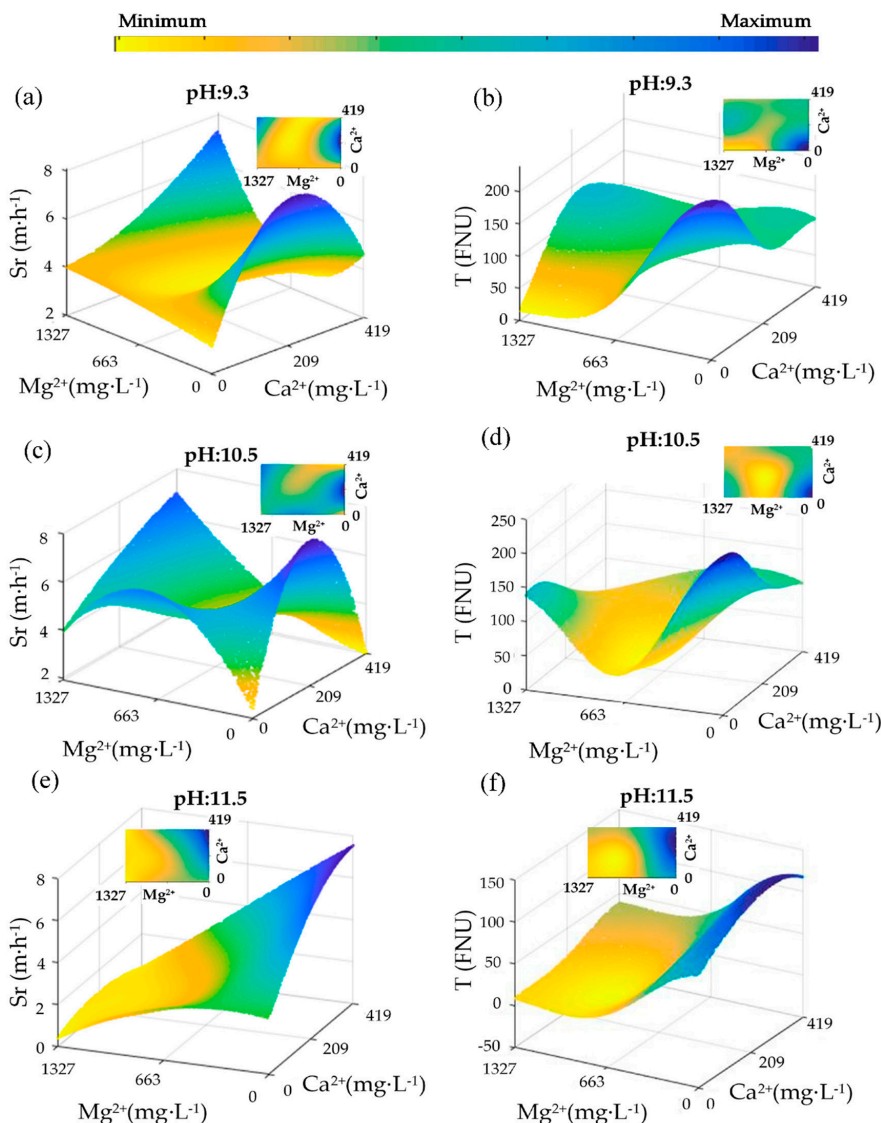

**Figure 10.** Response surface method (RSM) analysis from data elaborated by the Radial Basis Function Network (RBFN) model for Sr and T as functions of $Ca^{2+}$ and $Mg^{2+}$ concentrations at different pH: (**a**,**b**) for pH 9.3; (**c**,**d**) for pH 10.5; (**e**,**f**) for pH 11.5.

Figure 10a shows two Sr maximum points at pH 9.3: the highest corresponds to concentrations of $Ca^{2+}$ = 205–254 and $Mg^{2+}$ = 1–27 mg·$L^{-1}$, whereas the second shows $Ca^{2+}$ = 401–418 and $Mg^{2+}$ = 1295–1326 mg·$L^{-1}$. At this pH, most of the ions are dissolved, and some chemical complexes like $Ca(OH)^+$ and $Mg(OH)^+$ can be formed. According to the literature, these species would limit the volume of the polyacrylamide in the solutions. However, the play between polymer adsorption and its size in solution means that the sedimentation responses are not monotonous concerning the concentration of each ion. Turbidity is reported in Figure 10b,d, and only a point of minimum is present for concentrations of $Ca^{2+}$ = 0–30 mg·$L^{-1}$ and $Mg^{2+}$ = 1167–1324 mg·$L^{-1}$.

At pH = 10.5 in Figure 10c, it can be noticed three points of the maximum value for Sr; two of them coincide with the two peaks indicated for pH = 9.3, whereas the third point is placed in correspondence of concentrations of $Ca^{2+}$ = 1–12 $mg·L^{-1}$ and $Mg^{2+}$ = 608–875 $mg·L^{-1}$. At pH = 10.5, the mine tailings settling rates are, on average, higher than those at pH=9.3. This aspect is due to the different speciation of $Ca^{2+}$ and $Mg^{2+}$ ions induced by different values of pH. At pH = 9.3, $Ca^{2+}$ and $Mg^{2+}$ ions mainly are dissolved in an ionized state or form positively charged complexes (i.e., $Ca(OH)^+$ and $Mg(OH)^+$). In both cases, the presence of positive charges on their surface drives them to be adsorbed by the negatively charged polymer flocculant in competition with colloidal solids or also be adsorbed by colloidal solids and stabilize them. For this condition of pH (i.e., 10.5), the area of minimum turbidity is for concentrations of $Ca^{2+}$ lower than 251 $mg·L^{-1}$ and $Mg^{2+}$ between 400 and 800 $mg·L^{-1}$ (Figure 10d).

Figure 10e for pH = 11.5 shows a unique maximum peak of Sr at the highest $Ca^{2+}$ and lowest $Mg^{2+}$ concentration simultaneously. In these conditions of pH, mine tailings settling rate and supernatant turbidity are, on average, both lower than those evaluated at pH = 9.3 and pH = 10.5. Increased $Mg^{2+}$ removal leads to a higher sedimentation rate, while $Ca^{2+}$ plays a beneficial role within the concentration range. With this, it is suggested that the effort in the treatment of seawater be focused on removing $Mg^{2+}$ ions to reduce the presence of solid precipitates, but $Ca^{2+}$ does not show harmful effects. However, it is necessary to be careful with the selected treatment, since Jeldres et al. [16] used lime to form the precipitates, which were subsequently removed by vacuum filtration. Although the authors focused on reducing the magnesium content, the remaining calcium concentration was over 2000 ppm. Until now, it is unknown if such a concentration level impacts on the efficiency of the process. In general, the turbidity of the supernatant (Figure 10f) is lower than that obtained at pH 9.3 (Figure 10b) and 10.5 (Figure 10d), especially the region with the highest magnesium concentration. The lower sedimentation rate leads to better quality clarification.

Table 7 summarizes the range of maximum values of Sr and the relative areas of $Ca^{2+}$ and $Mg^{2+}$ concentrations at different pH. In contrast, Table 8 summarizes the range of minimum values of T and the relative areas of $Ca^{2+}$ and $Mg^{2+}$ concentrations at different pH. The values of both tables result from the analysis of 3D surfaces in Figure 10.

**Table 7.** Optimal concentration ranges of $Ca^{2+}$ and $Mg^2$ for the highest mine tailings settling rate (Sr) as a function of pH.

| pH | Settling Rate ($m·h^{-1}$) | $Ca^{2+}$ ($mg·L^{-1}$) | $Mg^{2+}$ ($mg·L^{-1}$) |
|---|---|---|---|
| 9.3 | 7.92–8.08 | 205–254 | 1–27 |
|  | 7.04–7.31 | 401–418 | 1295–1326 |
| 10.5 | 6.85–6.97 | 182–224 | 3–24 |
|  | 6.38–6.47 | 386–418 | 1286–1326 |
|  | 6.07–6.15 | 1–12 | 608–875 |
| 11.5 | 7.05–7.52 | 387–418 | 3–90 |

**Table 8.** Optimal concentration ranges of $Ca^{2+}$ and $Mg^2$ for the lowest residual turbidity (T) as a function of pH.

| pH | Turbidity (FNU) | $Ca^{2+}$ ($mg·L^{-1}$) | $Mg^{2+}$ ($mg·L^{-1}$) |
|---|---|---|---|
| 9.3 | 15.55–34.30 | 0–30 | 1167–1324 |
| 10.5 | 25.19–38.72 | 397–416 | 1–72 |
|  | 34.56–39.12 | 118–201 | 597–796 |
| 11.5 | 0–10 | 80–226 | 796–1128 |

Table 9 shows, at different pH, the optimal ranges of $Ca^{2+}$ and $Mg^{2+}$ concentration for optimizing, simultaneously, settling rate and turbidity. To find these optimums, an analysis of the 3D graphics was

carried out by selecting areas that have high Sr and low T. In this case, the peaks are areas that meet the requirements (Table 9).

**Table 9.** Concentration ranges of $Ca^{2+}$ and $Mg^2$ for optimizing the sedimentation process performance at pH of 9.3, 10.5, and 11.5.

| pH | $Ca^{2+}$ (mg·L$^{-1}$) | $Mg^{2+}$ (mg·L$^{-1}$) | Settling Rate (m·h$^{-1}$) | Turbidity (FNU) |
|---|---|---|---|---|
| 9.3 | 169–338 | 0–130 | 5.74–8.07 | 103.8–143.6 |
| 10.5 | 0–21 | 400–741 | 5.73–6.15 | 44.54–63.07 |
| 11.5 | 377–418 | 703–849 | 2.6–3.7 | 32.8–44.65 |

*3.5. Sedimentation Tests in Chemically Pretreated Seawater (CHSw), Pretreated Seawater by Biomineralization (BSw), Tap Water (Tw) and Raw Seawater (Sw)*

Figure 11 reports the response of five parameters (mine tailings settling rate, turbidity, Cp, % WR, and 10 M NaOH solution volume consumption) for each aqueous medium investigated and three different pH (9.3, 10.5, and 11.5). For instance, in Figure 11a, the values of Sr resulting from four sedimentation tests are reported.

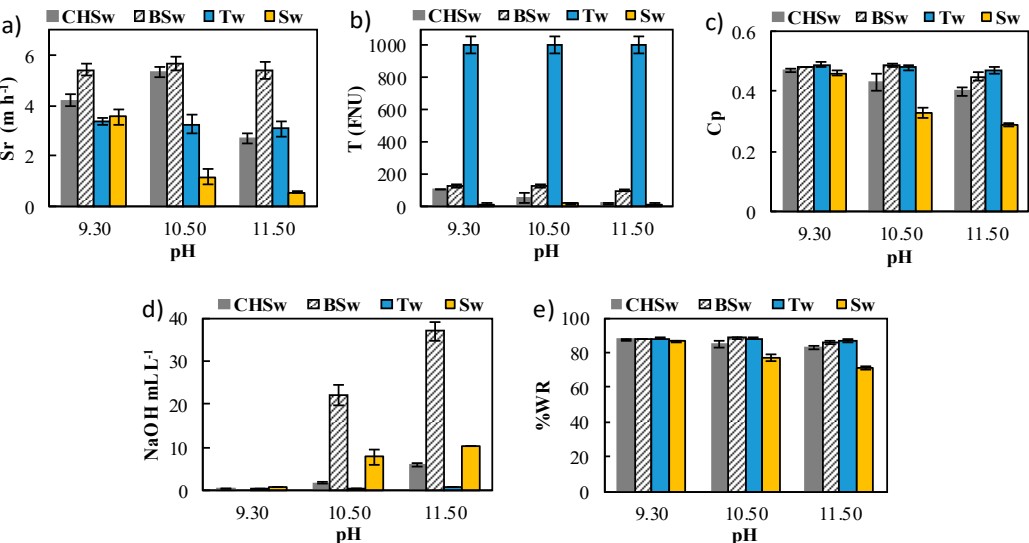

**Figure 11.** Process performance parameters (**a**) Settling rate (Sr), (**b**) Turbidity (T), (**c**) Solid fraction (Cp), (**d**) NaOH consumption, (**e**) Water recovery percentage (%WR) in sedimentation tests and different aqueous media (i.e., Chemically pretreated seawater (CHSw), Pretreated seawater by biomineralization (BSw), Tap water (Tw) and Raw Seawater (Sw) at different pH values (i.e., 9.3, 10.5, and 11.5).

In this figure, it can be noticed that the highest Sr values (i.e., 5.4, 5.7, and 5.4 m·h$^{-1}$) were reached independently of pH when BSw was used as the separation medium. The ANOVA analysis indicates significant differences between means of one type of water compared with the others when the Sr is evaluated, thus, obtaining the best results in BSw. This result is reasonable due to the presence of a high concentration of $NH_4^+$ ion ranging from 4000 and 7000 mg·L$^{-1}$ in the liquid bulk used to produce BSw. The $NH_4^+$ concentration in the medium is due to urea hydrolysis by urease enzyme from *B. subtilis* LN8B. Per mole of urea, urease enzyme hydrolyzes urea into one mole of carbonate ion and two moles of ammonium ion. Carbonate ions precipitate as calcium carbonate in the presence of calcium ions from seawater, decreasing its concentration as a soluble ion [18]. On the other hand, $NH_4^+$ ions can remain in solution or form precipitates as struvite or another crystal, helping to reduce the soluble concentration of $Mg^{2+}$ from seawater [36]. Soluble ammonium reacting with OH$^-$ dissolved in water forms $NH_3^+$ and, at the same, reduces the concentration of OH$^-$ that, otherwise, would react with $Ca^{2+}$ and $Mg^{2+}$ to form solid complexes that reduce the performance of polyacrylamides (see Section 3.2).

Focusing the attention on BSw, it can be noticed that residual turbidity (Figure 11b), as well as Cp (Figure 11c) and % WR (Figure 11e), are not significantly affected by pH. A different conclusion is drawn when 10 M NaOH solution is considered: actually, this parameter shows a proportional dependence on pH; this result is expected as NaOH is used to regulate the pH. For CHSw, the highest mine tailings settling rate is reached at pH 10.5; the turbidity decreases with an inverse proportion tendency with pH, even if all results are remarkable, because the residual turbidity is less than 100 FNU; Cp fluctuates between 0.4 and 0.5; % WR ranges between 0.80 and 0.90; NaOH solution is used to modify pH, thus, showing a trend similar to BSw but less outstanding. Sedimentation performance in Tw shows a result independent of pH: the settling rate is approximately constant and equal to 3 m·h$^{-1}$; residual turbidity is the highest at any pH (i.e., over 1000 FNU), thus, proving that salinity promotes the aggregation of colloidal solids; Cp and % WR do not vary significantly with pH and show values similar to those resulting from sedimentation tests in BSw; NaOH consumption to regulate the pH is small, as Tw does not have a buffer effect. Finally, the sedimentation process in Sw is the poorest performing except for residual turbidity: actually, the mine tailings settling rate ranges between 0.5 and 3.6 m·h$^{-1}$, decreasing with pH; residual turbidity is the lowest (i.e., less than 20FNU), which agrees with previous works [37]; Cp and % WR do not vary significantly, even if an inverse dependence on pH is noticed; the consumption of NaOH, as already discussed for BSw and CHSw, increases with pH. Interestingly, Tw represented the worst performing aqueous medium to conduct the sedimentation process, as tests carried out with it resulted in the highest residual turbidity coupled with a modest mine tailings settling rate.

Table 10 shows the comparison between experimental data resulting from the sedimentation tests conducted in CHSw, BSw, Tw, and Sw with those predicted by the RBFN model that were calibrated with experiments in ASw. It is noticed that there is a high variability between experimental and predicted results for sedimentation tests conducted in BSw, Tw, and Sw. It is highlighted that the effectiveness of CHSw is well predicted by the RBFN model, whereas the effectiveness of BSw is not affected by pH, and those of Tw and Sw are overestimated, since the T for Tw was higher than that predicted, and T for Sw was lower than that predicted.

**Table 10.** Comparison between experimental data and those predicted by the Radial Basis Function Network model.

| | Independent Parameters | | | Response Parameters | | | |
|---|---|---|---|---|---|---|---|
| | | | | Settling Rate (m·h$^{-1}$) | | Turbidity [FNU] | |
| | Ca mg·L$^{-1}$ | Mg mg·L$^{-1}$ | pH | Experimental Data | RBFN Data | Experimental Data | RBFN Data |
| CHSw | 263 | 648 | 9.3 | 4.207 | 3.19 | 105.75 | 89.1 |
| | | | 10.5 | 5.344 | 4.76 | 51.375 | 44.25 |
| | | | 11.5 | 2.673 | 2.28 | 18.142 | 25.3 |
| BSw | 5.8 | 284 | 9.3 | 5.432 | 3.5 | 126 | 186 |
| | | | 10.5 | 5.659 | 5.35 | 129 | 145 |
| | | | 11.5 | 5.420 | 2.84 | 100 | 62.54 |
| Tw | 75 | 26 | 9.3 | 3.375 | 5.68 | 1000 | 213.34 |
| | | | 10.5 | 3.264 | 5.73 | 1000 | 209.58 |
| | | | 11.5 | 3.072 | 4.08 | 1000 | 103.92 |
| Sw | 395 | 1270 | 9.3 | 3.552 | 6.95 | 15 | 90.98 |
| | | | 10.5 | 1.165 | 5.96 | 18 | 55.77 |
| | | | 11.5 | 0.553 | 0.73 | 14.7 | 37.51 |

## 4. Conclusions

This work has investigated, by the use of RBFN methodology, the best chemical composition of partial desalinated seawater, in terms of $Ca^{2+}$ and $Mg^{2+}$ concentration, to optimize the sedimentation process performance. It was found that the optimal ranges of $Ca^{2+}$ and $Mg^{2+}$ concentration are respectively as follows: 169–338 and 0–130 mg·L$^{-1}$ at pH 9.3; (ii) 0–21 and 400–741 mg·L$^{-1}$ at pH 10.5; (iii) 377–418 and 703–849 mg·L$^{-1}$ at pH 11.5. Furthermore, this study has proved that partial desalinized seawater is more efficient than raw seawater as well as tap water in conducting the



sedimentation process for mining purposes. Such results can have a highly positive impact on the environment without depressing the economy, as mining factories could stop using freshwater as well as desalinated seawater to process ores, thus, reducing significantly the environmental as well as financial costs. Partial desalinized seawater needs less production costs than totally desalinated water, even less if pretreated seawater by biomineralization methods is preferred to chemical. Moreover, a partial desalinized water is better performing than seawater for the sedimentation process and causes less damages to pipelines and other devices used for mining purposes. Finally, partially desalinated water could replace completely freshwater currently used by mining factories, thus, preserving this strategic resource.

**Author Contributions:** G.V. performed the experiments, modeling, analysis of results and writing this article; D.A. participated in devising the presented idea and writing; R.J. analyzed the results of sedimentation assays and writing; A.P. participated in writing this article and editing the English language; M.R. participated in the design of the trials, reviewing this article, and funding acquisition; L.A.C. participated in project administration, modeling, funding acquisition, designing the experiments, and writing this article. All authors have read and agreed to the published version of the manuscript.

**Funding:** The authors thanks the support of ANID through Anillo–Grant no. ACM 170005.

**Conflicts of Interest:** The authors declare no conflict of interest.

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
