# Peer review of "Use of Radial Basis Function Network to Predict Optimum Calcium and Magnesium Levels in Seawater and Application of Pretreated Seawater by Biomineralization as Crucial Tools to Improve Copper Tailings Flocculation"

_minerals, doi:10.3390/min10080676_

Round 1

Reviewer 1 Report

I have one central problem and that is the authors did not justify the use of neural networks. How did the results different from linear models, e.g. multiple linear regression model!

Author Response

Dear Reviewer

Thanks

Dayana

Reviewer 2 Report

Dear authors, dear editor.

I have read the manuscript titled “Neural networks and pretreated seawater by biomineralization as alternatives to improve copper tailings flocculation”. In this paper authors made an experiment where they investigated the effects of Ca, Mg concentrations and pH to the settling rate and turbidity, in the case of producing ore concentrates by use of sea water and flocculants.

The study is interesting and relevant for the readers of Minerals journal, so I can recommend publishing. However, there are some remarks which needed to be addressed prior publication.

  • Title: please be more specific, current title is not appropriate. Perhaps may I suggest new title: “use of radial basis function network to predict optimum Ca and Mg levels and pH to optimize sedimentation rate when producing Cu concentrate by the use of sea water”
  • Introduction: please add a paragraph explaining the current production process of ore concentrate for your case
  • L46 – L98: this section could be shortened. Please focus only on providing an information relevant for your case.
  • L185: Much more info is needed at that point: RBFN topology, data preparation, learning methods, validation and other info which will allow reconstruction of your study by other researchers. Also I have another question at that point, which needs to be addressed in the text: i) why using FBFN and why not other neural networks (i.e. multilayer perceptron, self-organising maps and similar), which are more commonly used today as RBFN; ii) why particularly you use neural networks, and not other more "conventional" modelling techniques, because your process is not very complicated (only 3 independent and 2 dependant variables), iii) the amount of data you used for learning is very low. This is a serious drawback of your study, which also does not allows to properly evaluate the results (see also one of my future comment), iv) I am asking myself - why do you need RBFS at all in your study?
  • Consider presenting your results in a form of three-component diagrams. It will be much easier to understand and visualize. And with such diagram, when interpolated with generally used interpolation tools, you do not need RBFN at all. Did you extrapolate your results?
  • L259: How do you know your process is non-linear?
  • Table 5 and figure 5: Artificial neural networks (ANNs) are universal approximators. If the network is large enough you can easily obtain 100% accuracy. But this does not means the network has any predictive power, because overfitting is a well-known problem at using ANNs. You must validate your network with real data which was not included in a learning dataset, but this could be a problem in your case because the amount of data you have is quite low. So this validation is pointless.
  • Figure 6: You do not need RBFS to construct such models. You can use simple linear, exponential, power or polynomial fitting, or maybe even other interpolation techniques (like kriging, natural neighbour etc.).
  • Table 9: you can use part of this data for learning, and another part for validation? It might improve your results.
  • Please make sure that all abbreviations are properly explained. I suggest explaining them also in connection with the tables / figures (in captions), because readers might not want to read the whole text to find a meaning for a specific abbreviation.
  • Although I am not native English speaker, I can spot several flaws in the use of language and grammar. Please use native speaker to address this issue.

I hope you will find my review useful. Good luck with the revision. Best regards.

Author Response

Dear Reviewer

Thanks

Dayana Arias

Round 2

Reviewer 1 Report

Revisions are acceptable. 

Author Response

Thank you. Please see file enclosed

Reviewer 2 Report

Dear authors, dear editor.

I have read revised version of the manuscript titled “Use of radial basis function network to predict optimum calcium and magnesium levels in seawater and application of pretreated seawater by biomineralization as alternatives crucial tools to improve copper tailings flocculation”.

Unfortunately I can report that my previous comments have not been addressed in sufficient manner, especially, because corrections are too general and does not allow the reader to reconstruct your work. In particular:

  • Description of Cu ore processing is too general and certainly does not tell anything about the process for your specific case
  • Description of RBFN approach to modelling is also general and not specific for your study (for example: how many hidden neurons you used, what was the learning algorithm and what were the parameters etc.)
  • Reasoning why RBFN can be used for solving your problem is still not sufficiently addressed and is very weak
  • Model performance control has not been improved and remains a weak point of this study

Best regards.

Author Response

Dear Reviewer

We want to thank you for spending valuable time reading and giving insightful comments on our paper. All the comments were addressed, as it is indicated below.

Comments and Suggestions

  • Description of Cu ore processing is too general and certainly does not tell anything about the process for your specific case

We are not working with a particular ore or process. We use an artificial tailing of 80% w/w quartz and 20 % w/w kaolinite. This type of mixture may have several applications and is not related to a particular mineral. However, a brief description of the effect of seawater on mineral flotation was included. Also, new references were added. The paragraph is described below, and it was included in  the ‘Introduction’ Section (L44-L48):

for example, in flotation operations affect bubble stability, activation of minerals, pulp rheology, among others. There are several studies and reviews about the use of raw seawater directly in flotation processes [5–10]. A review of these studies is outside of the objective of this manuscript. However, another effect in mining operations includes low tailings thickening efficiency when operating under highly alkaline conditions

  • Description of RBFN approach to modelling is also general and not specific for your study (for example: how many hidden neurons you used, what was the learning algorithm and what were the parameters etc.)

In section 2.7 a description of the RBFN method was given as follow:

2.7. Radial Basis Function Network (RBFN)

Neural networks can represent non-linear assignments of multiple inputs to one or more outputs. These can also be applied to regression problems. The neural network is capable of mapping input variables to continuous values. An important class of neural networks is RBFN. The RBFN consists of an input layer, a hidden layer (Radial Basis Layer), and an output layer (linear layer). The structure of RBFN is presented in Figure 3, where the inputs are represented by p1 and the outputs by a2.

In the Radial Basis Layer, the distance between the input vector pi and the rows of the weight wi1 are calculated. After that, bias bi multiply it, so in this way the net input for the neuron i () in the Radial Basis Layer is calculated by Equation 4:

  (4)

Where bi is the bias which is related with standard deviation σ with equation 5.

(5)

The transfer function used in the Radial Basis Layer or hidden layer is a Gaussian function, which is commonly used in the neural network community. Equation 6 and Figure 2 represent it.

(6)

Figure 2 Gaussian Basis function

Finally, the linear layer or the output layer is represented by Equation 7:

(7)

Figure 3 illustrates the RBFN structure and its equations associated.

Figure 3 Radial Basis Function Network (RBFN) structure

The performance of the RBFN is evaluated using the mean squared error (MSE). The next equations describe the sequence to calculate the MSE using the RBFN. Let consider the following training points, Equation 8:

(8)

The neuron in the hidden layer is calculated by equations 9 and 10                                                                                                                                       

(9)

(10)

Grouping terms, the following points are obtained:                                                                                                                                       

(11)

Then the response a2 is determined in the output layer through equation 12,                                                                                                                                       

(12)

Finally, the MSE performance index is determined using Equation 13 for the training set [23]:                                                                                                                                        

(13)

The newrb function of MatLab was used for modeling the sedimentation rate and turbidity using RBFN. Two RBFN networks were built, one for the sedimentation rate and another for the turbidity. Both networks have three input variables (Ca2+ and Mg2+ concentrations and pH). The newrb function works with the following code: net = newrb (p, t, GOAL, SPREAD). The p matrix represents the input variables and t the output variables. GOAL is the target error, SPREAD represents the spread width, which can take different values that will influence the MSE. The network is developed by adding one neuron at a time, and each calculates the MSE. In the iterative process, the MSE value is compared with the GOAL, and if the MSM is equal to or less than GOAL then the process converges; otherwise, another neuron is added and so on [24]. This iterative process is performed for various values of SPREAD.

- On the other hand, the results of the application of RBFN is described in section 3.2 as follow:

3.2. Analysis of Radial Basis Function Network (RBFN) model

Artificial neural networks allow experimental data to be processed, generating a predictive data set (RBFN output data) which is suitable for achieving the optimal solution of non-linear equations [33]. After calibration, the RBFN model turns to be a potent tool capable of predicting, with accuracy, the output response variables from any combination of input values (independent variables). Recall that the RBFN structure has an input layer, a hidden layer, and an output layer. Within our case, the RBFN has three input variables (Ca2+ concentration, Mg2+ concentration, and pH) and two output variables (Sedimentation Rate, Sr, and Turbidity, T). The MatLab computer program was used to found the optimal number of neurons and the width (w) described in Section 2.7. The objective function was to minimize MSE [34].

In data processing with RBFN, 27 data points were processed for settling rate and turbidity. In the first step, the three input variables were normalized to values in the range [0,1] to facilitate the calculation and eliminate the influence of dimension. In the second step, the function newrb (p,t,GOAL,SPREAD) of MatLab was applied with a GOAL of 0.05. Several values of SPREAD were employed, and SPREAD values of 2.49 and 0.4 were found for the settling rate and turbidity, respectively. For these SPREAD values, 20 and 25 neurons were gotten for settling rate and turbidity networks. Figure 7 provides the RBFN structures obtained. Table 5 presents weight w and bias b values for hidden (w1, b1) and output layer (w2, b2) for settling rate and turbidity, respectively.

Table 5. Settling rate and Turbidity weights w and bias b values from processing data by MatLab.

Settling rate Sr

Turbidity T

w1

w2T

b1

b2

w1

w2T

b1

b2

1

0

0

160320.7

0.3344

-53.425

0

0

0.5455

174.2

2.0814

-2.622

0.4999

1

1

583055.1

1

0

0.5455

-10.8

1

0.5

0.5455

400102.9

0

0

0

198.1

1

1

1

-227975.1

0.4999

1

0

123.2

0

0.5

1

386413.9

0

1

0.5455

196.2

0

0

1

-126579.5

0.4999

0

1

71.7

0

1

1

-245670.7

1

0.5

0

8.9

0.4999

0

0.5455

232292.3

0

0.5

0.5455

-88.9

1

0.5

1

252277.5

0

1

0

-76.5

0.4999

0

1

256532.2

0

1

1

-67.4

0

1

0

17756.8

1

1

1

43.5

0.4999

1

0.5455

-141222.4

1

0

0

92.4

1

0

0.5455

-380104.8

1

0

1

90.2

0

0

0.5455

-107365.3

1

0.5

0.5455

99.9

1

0.5

0

-223612.6

0.4999

0.5

1

-14.4

1

1

0.5455

-83433.9

0.4999

0.5

0.5455

-75.0

0

0.5

0.5455

64578.8

0.4999

0

0.5455

56.4

0.4999

0.5

1

-870233.9

1

0.5

1

-28.2

0.4999

0

0

-39731.5

0.4999

1

0.5455

33.6

1

1

0

92488.7

0

0.5

1

30.6

1

1

0

28.5

1

1

0.5455

-23.2

0

0

1

-12.6

0.4999

0.5

0

19.1

0.4999

0

0

-10.6

Figure 7. MatLab structure for RBFN a) Settling rate and b) Turbidity

According to values shown in Table 6, the statistical parameter R2 resulting from the calibration process was 0.98 and 0.99 for Sr and T, respectively. These values, coupled with other statistical parameters reported in Table 6, corroborate the reliability of the RBFN model in simulating results from the sedimentation process. Figure 8 shows graphically the fitting between experimental data and those predicted by the RBFN model.

Table 6. Statistical parameters resulting from the calibration of the Radial Basis Function Network (RBFN) model.

Statistical parameters

Value

Sr prediction

T prediction

Mean relative error

0.1104

0.00597

Mean square error

0.1638

0.29917

Root mean square

0.4047

0.54697

R squared

0.9809

0.99996

Adjusted R squared

0.9774

0.99995

Figure 8. Comparison between experimental and predicted by Radial Basis Function Network (RBNF) model data.

  • Reasoning why RBFN can be used for solving your problem is still not sufficiently addressed and is very weak

As mentioned previously, the traditional models of adjustment by the multiple regression method or others do not represent the data accurately. If we look at the results of data adjustments in section 3.1, the adjustment R2 for the traditional method is 14 and 35% for the Sedimentation rate Sr and Turbidity T, respectively. Therefore, another method is needed, such as RBFN. The advantage of using RBFN method is described in the last part of section 2.7 as follow (L231-237):

It is worth indicating that RBFN was selected because a recent study shows that RBFN performs better than multi-layer perceptron networks when it is applied to response surface methodology [25]. Another advantage of RBFN over other neural networks is that during RBFN learning, the inputs are fed directly from the hidden layer without any weight, and the weights are only manifested between the hidden and the output layer. Such weights are modified depending on the error. Thus, RBFN requires a much shorter learning time compared to multi-layer feedback neural networks, used in other practical applications, and therefore, the convergence time is also considerably short [26]

Also, in section 3.1 the following was added:

Several multiphase systems do not follow a second-order polynomial behavior how was observed here. The immediate consequence is incorrect optimization. In these cases, the most popular alternative is to use ANNs [28]. For this reason, the RBFN method was chosen, and it was advantageous to show more comprehensively the results (see section 3.2).

  • Model performance control has not been improved and remains a weak point of this study

In section 3.2, a better description of RBFN was included. In addition, section 3.3 shows the validation, which gives consistent results. This is described in the new manuscript version as follows:

3.3. Radial Basis Function Network (RBFN) validation

There are different methods of network validation. These methods mainly depend on the amount of data available. According to Kopal et al. [35], three ways of validation are known. For small data set, the following validation method is adequate. The idea of this method is to train the network several times using the following methodology: from a set composed of a number n of samples or points, one of them is omitted, and the neural network is trained, then the next point is omitted, and the neural network is re-trained, so one point at a time is removed, and the network is trained. This process is iteratively performed as long as n subsets of respond data are generated. One of them is left for validation; therefore, n-1 sets of responses are used to calibrate the model, and the remaining one is used for its validation.

The validation results are displayed in Figure 9, where the calculated values (average data) are compared with the last 27th set of data (validation). According to the previously described procedure, the validation for the Settling rate (Sr) and Turbidity (T) have adjusted R2 of 0.90 and 0.92, respectively.

Figure 9. Comparison between validation and average data in the validation process